# LIFR inhibition enhances the therapeutic efficacy of HDAC inhibitors in triple negative breast cancer

Mengxing Li[1,2,13], Suryavathi Viswanadhapalli [1,13✉], Bindu Santhamma[3], Uday P. Pratap[1], Yiliao Luo[1,4], Junhao Liu[1,5], Kristin A. Altwegg[1,6], Weiwei Tang[1,7], Zexuan Liu[1,5], Xiaonan Li[1], Behnam Ebrahimi[1], Hui Yan[8], Yi Zou[9], Swapna Konda[3], Gangadhara R. Sareddy [1,6], Zhenming Xu[6,8], Yidong Chen[6,9], Manjeet K. Rao [6,9], Andrew J. Brenner[6,10], Virginia G. Kaklamani[6], Rajeshwar R. Tekmal[1,6], Gulzar Ahmed[3], Ganesh V. Raj [11], Klaus J. Nickisch[3], Hareesh B. Nair [3✉] & Ratna K. Vadlamudi [1,6,12✉]

Histone deacetylase inhibitors (HDACi) are identified as novel therapeutic agents, however, recent clinical studies suggested that they are marginally effective in treating triple negative breast cancer (TNBC). Here, we show that first-in-class Leukemia Inhibitory Factor Receptor (LIFRα) inhibitor EC359 could enhance the therapeutic efficacy of HDACi against TNBC. We observed that both targeted knockdown of LIFR with CRISPR or treatment with EC359 enhanced the potency of four different HDACi in reducing cell viability, cell survival, and enhanced apoptosis compared to monotherapy in TNBC cells. RNA-seq studies demonstrated oncogenic/survival signaling pathways activated by HDACi were attenuated by the EC359 + HDACi therapy. Importantly, combination therapy potently inhibited the growth of TNBC patient derived explants, cell derived xenografts and patient-derived xenografts in vivo. Collectively, our results suggest that targeted inhibition of LIFR can enhance the therapeutic efficacy of HDACi in TNBC.

[1] Department of Obstetrics and Gynecology, University of Texas Health San Antonio, San Antonio, TX 78229, USA. [2] Department of Respiratory Medicine, Xiangya Hospital, Central South University, Hunan 410008, P.R. China. [3] Evestra, Inc, San Antonio, TX 78245, USA. [4] Department of General Surgery, Xiangya Hospital, Central South University, Hunan 410008, P.R. China. [5] Department of Oncology, Xiangya Hospital, Central South University, Hunan 410008, P.R. China. [6] Mays Cancer Center, University of Texas Health San Antonio, San Antonio, TX 78229, USA. [7] Department of Obstetrics and Gynecology, Affiliated Hospital of Integrated Traditional Chinese and Western Medicine, Nanjing University of Chinese Medicine, Nanjing 210028, China. [8] Department of Microbiology, Immunology and Molecular Genetics, University of Texas Health San Antonio, San Antonio, TX 78229, USA. [9] Greehey Children's Cancer Research Institute, University of Texas Health San Antonio, San Antonio, TX 78229, USA. [10] Department of Hematology & Oncology, University of Texas Health San Antonio, San Antonio, TX 78229, USA. [11] Departments of Urology and Pharmacology, University of Texas Southwestern Medical Center at Dallas, Dallas, TX 75390, USA. [12] Audie L. Murphy Division, South Texas Veterans Health Care System, San Antonio, TX 78229, USA. [13]These authors contributed equally: Mengxing Li, Suryavathi Viswanadhapalli. ✉email: viswanadhapa@uthscsa.edu; hnair@evestra.com; vadlamudi@uthscsa.edu

Breast cancer (BC) is the most common cancer in women with an estimated 281,550 new cases and about 43,600 women are expected to die in 2021 from BC in the U.S. (American Cancer Society, Cancer Facts & Figures 2021). Among the different subtypes of BC, 15–24% are triple-negative breast cancer (TNBC)[1]. TNBC is more aggressive, and due to limited targeted therapies, represents a disproportional share of the BC mortality[2,3]. There is a critical need for rationally designed therapeutics that can improve response to TNBC treatment.

Epigenetic changes are implicated in the progression of many cancers including TNBC[4–6]. Histone acetyltransferases (HATs) and histone deacetylases (HDACs) determine the acetylation status of histones. HATs and HDACs affect gene expression; and inhibitors of HDACs (HDACi) cause growth arrest, differentiation, and/or apoptosis of many cancers[7]. The FDA has approved histone deacetylase inhibitors (HDACi), a class of small-molecular therapeutics as anticancer agents for various cancers including T-cell lymphoma and multiple myeloma. HDACi, such as vorinostat, romidepsin, and panobinostat are being tested for treating TNBC in clinical trials. Although HDACi monotherapy showed good promise in treating hematological malignancies, they are marginally effective in treating solid tumors such as TNBC[5,8].

Leukemia inhibitory factor receptor (LIFR, also referred as LIFRα, LIF Receptor Subunit Alpha, CD118)[9] and its ligand LIF[10], are widely expressed in many solid tumors including BC[11], and their overexpression is often associated with poor prognosis for patients. Recent studies suggested LIF-LIFR axis as a promising therapeutic target for cancer therapy[12,13]. LIF binding to LIFR complex comprised of LIFR and glycoprotein 130 (gp130)[9], triggers LIFR signaling to activate multiple signaling pathways including JAK, STAT, MAPK, AKT, and mTOR[9,11,14], all of which are implicated in TNBC progression. Recent studies have elucidated that HDACi treatment promotes expression and activation of LIFR, which restrains the utility of HDACi for BC treatment[15]. Targeting LIFR may serve as a unique approach to enhance the efficacy of HDACi for treatment of TNBC. We have rationally designed, synthesized and selected a lead small organic molecule, EC359 that can emulate the LIF–LIFR binding site and, therefore, function as a LIFR inhibitor[16]. We hypothesized that EC359 will have utility in enhancing the efficacy of HDACi by inhibiting LIFR oncogenic signaling.

Here, we examined the therapeutic utility of EC359 in improving the efficacy of HDACi in TNBC models. Our results showed that multiple HDACi that are currently in clinical trials induce expression of LIFR which compromise the efficacy of HDACi treatment. Utilizing LIFR knockout cell lines, we provided genetic evidence on the essential role of LIFR in HDACi-mediated aberrant activation of oncogenic/survival signaling pathways. Using in vitro and in vivo models, we demonstrated the synergistic effect of EC359 + HDACi combination therapy. Mechanistically, EC359 inhibited HDACi induced LIFR oncogenic signaling pathways.

## Results

**HDACi treatment induces LIFR expression**. A recent study using whole-genome microarray discovered that HDACi (vorinostat) treatment induces expression of LIFR which reduces the efficacy of HDACi therapy[15]. We tested the generality of this finding by treating TNBC cells with four different HDACi (vorinostat, panobinostat, romidepsin, and givinostat) which are currently in clinical trials. Western blotting results using five different TNBC cells confirmed that treatment with all four HDACi indeed induced expression of LIFR (Fig. 1a and Supplementary Data Fig. 1a). RT-qPCR assays confirmed upregulation of LIFR mRNA expression (Fig. 1b). Further, HDACi induced LIFR expression is functional as measured by its downstream activation of STAT3 phosphorylation (Fig. 1a and Supplementary Fig. 1a) and STAT3-Luc reporter activity (Fig. 1c). We next determined the expression of LIFR ligands in TNBC cells. Western blot analysis confirmed expression of LIF in TNBC model cells (Fig. 1d). Further, RT-qPCR assays showed that TNBC cells also express other LIFR ligands, specifically OSM and CNTF (Supplementary Fig. 1b). In addition, gp130 which functions as a co-receptor of LIFR is also expressed in TNBC cells (Supplementary Fig. 1b) and HDACi treatment did not alter its expression (Supplementary Fig. 1c). Collectively, these findings confirmed that HDACi aberrantly activates the expression of LIFR and abundant expression of LIFR ligands in TNBC cells may contribute to autocrine signaling via LIFR.

**HDACi-mediated STAT3 activation requires the presence of LIFR**. To confirm the direct role of LIFR in HDACi-mediated activation of STAT3, we knocked out (KO) the expression of LIFR using the CRISPR/Cas9 system and then treated cells with HDACi. LIFR-KO substantially reduced the induction of STAT3 phosphorylation mediated by HDACi (Fig. 1d). In MTT cell viability assays, LIFR-KO increased the potency of HDACi in reducing cell viability (Fig. 1e). Further, in colony formation assays, LIFR-KO enhanced the ability of HDACi to reduce the cell survival of TNBC cells (Fig. 1f). Collectively, these results suggest that HDACi-mediated STAT3 activation requires the presence of LIFR.

**LIFR inhibitor EC359 synergizes with HDACi to decrease TNBC cell viability**. We examined whether treatment with EC359 enhances the therapeutic potency of HDACi on TNBC cells using MTT cell viability assays. As shown in Fig. 2a and Supplementary Fig. 2a, b, addition of EC359 enhanced the potency of HDACi in reducing cell viability compared to HDACi as monotherapy. Results showed that the combination index (CI) values were <1 in all the three HDACi combinations tested and confirmed that the combination therapy was synergistic. We also compared the utility of STAT3 inhibitor (NSC 74859) in combination with HDACi. Results indicated that STAT3 inhibitor mediated reduction in cell viability of TNBC cells requires μM concentrations and is not synergistic to HDACi (Supplementary Fig. 3a, b). On the contrary, EC359 at nM concentrations is synergistic to HDACi in reducing cell viability of TNBC cells.

**EC359 enhances potency of HDACi in biological assays in vitro**. We then evaluated the utility of EC359 + HDACi therapy using several biological assays. In clonogenic survival assays, EC359 enhanced HDACi ability to reduce the colony formation of MDA-MB-231 and BT-549 cells compared to monotherapy (Fig. 2b). Matrigel invasion assays demonstrated that combination therapy of EC359 + HDACi is more effective in reducing the invasion of MDA-MB-231 and BT-549 cells compared to monotherapy (Fig. 2c, d). We next examined the utility of EC359 + HDACi combination therapy in enhancing apoptosis using both Annexin V-PI and Caspase 3/7 assays. Results showed that EC359 increased HDACi ability to promote apoptosis in both assays (Fig. 2e). LIF-STAT3 axis is implicated in stem cell self-renewal and pluripotency[17]. Further, LIF signaling is implicated in the regulation of the transcription factors SOX2 and NANOG which play a critical role in stemness[18]. Analysis of TCGA databases revealed that high expression of LIFR and LIF correlates with the expression of mammary stemness gene set (Supplementary Fig. 4a). Treatment of CSCs with EC359 enhanced the ability of HDACi in reducing the cancer stem cells (CSCs) viability, and mammosphere formation (Supplementary

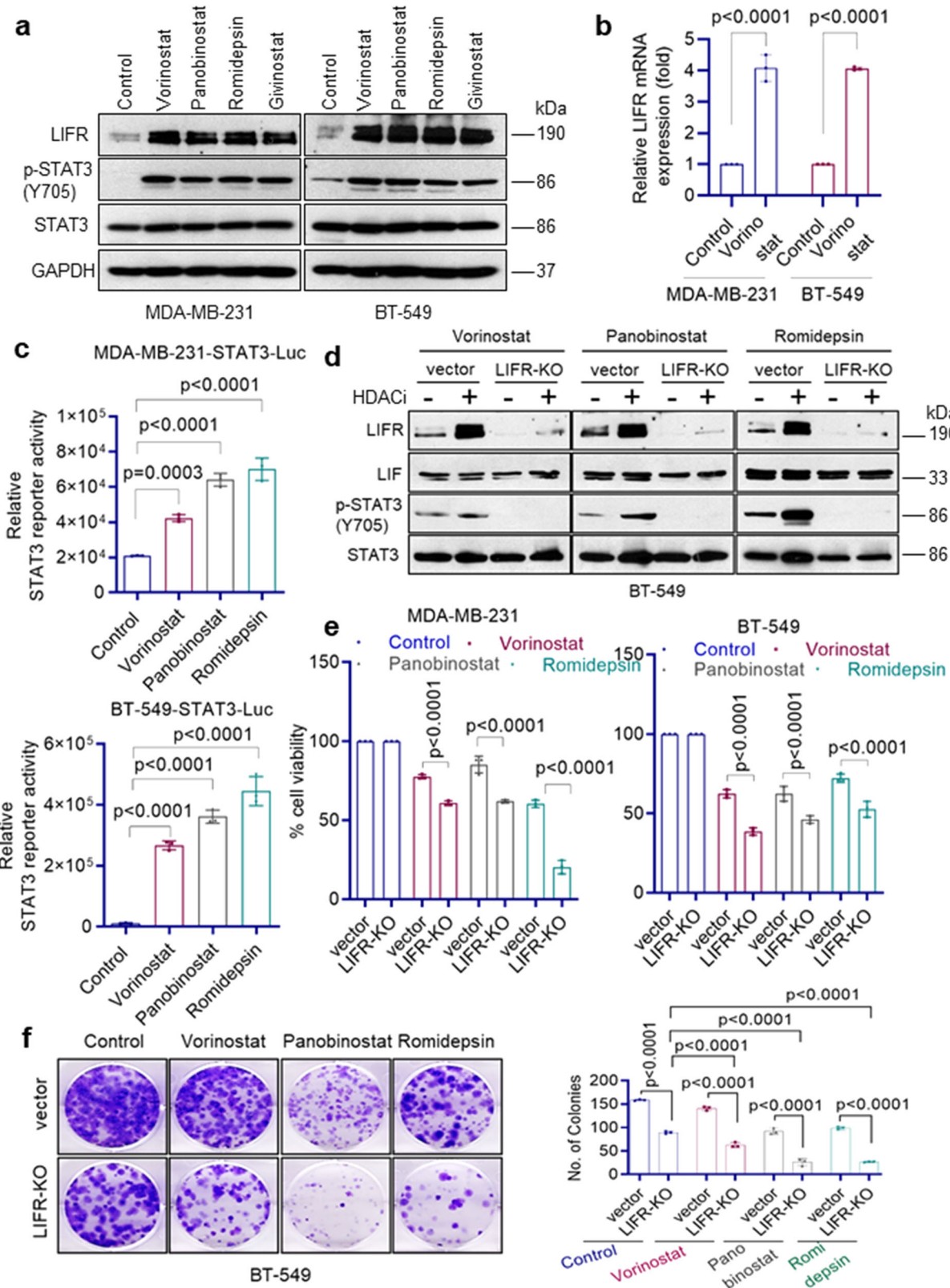

Fig. 4b-d). These results suggest that EC359 has potential to enhance the ability of HDACi to reduce cell survival, invasion, stemness, and enhance apoptosis of TNBC cells.

**LIFR expression is upregulated in TNBC tissues**. Recent studies showed that LIFR expression is upregulated during tumor progression of many cancers and contributes to therapy resistance[12,13]. Here, we examined whether LIFR expression is upregulated in BC using tissue microarrays that consist of various subtypes of BC and benign breast tissues. LIFR expression is higher in TNBC compared to benign, ER + and HER2 + BC (Fig. 3a). These results suggest that upregulation of LIFR expression occurs in TNBC.

**Fig. 1 HDACi treatment-induced STAT3 activation via LIFR expression. a** TNBC model cells (MDA-MB-231 and BT-549) were treated with indicated HDACi (vorinostat: 10 μM; panobinostat: 1 μM; romidepsin: 1 μM; givinostat: 1 μM) for 24 h and expression of LIFR, p-STAT3(Y705), and STAT3 were determined using Western blotting. **b** MDA-MB-231 and BT-549 cells were treated with vorinostat (10 μM) for 10 h and levels of LIFR were measured by RT-qPCR. RT-qPCR data were normalized to GAPDH and data are representative of three independent experiments (n = 3). **c** MDA-MB-231 and BT-549 cells stably expressing STAT3-luc reporter were treated with indicated HDACi and reporter activity was measured after 24 h. Data are representative of three independent experiments (n = 3). **d** BT-549 vec or BT-549 LIFR-KO cells were treated with indicated HDACi (vorinostat: 10 μM; panobinostat: 1 μM, romidepsin: 1 μM) for 10 h and induction of LIFR, LIF, and p-STAT3(Y705) was measured by Western blotting. The effect of LIFR-KO on the activity of HDACi was determined using MTT cell viability assay (**e**) and clonogenic survival assay (**f**). **e, f** Data are representative of three independent experiments (n = 3). Error bars represent SD. In **b, e**, and **f**, p-values were calculated using two-way ANOVA. In **c**, p-values were calculated using one-way ANOVA.

**EC359 enhances HDACi efficacy in xenograft assays.** To test the utility of EC359 + HDACi therapy in vivo, we conducted xenograft study using two different TNBC cells. MDA-MB-231 and MDA-MB-468 xenografts bearing SCID mice were randomized and treated with vehicle or EC359 (5 mg/kg/ip) 3 days/week or HDACi (vorinostat) (100 mg/kg/oral) 5 days/week alone or in combination. EC359 + HDACi combination treatment resulted in lower tumor volume and smaller tumor weight compared to monotherapy of EC359 or vorinostat (Fig. 3b, c). Mice treated with EC359 + HDACi combination exhibited no overt signs of toxicity. IHC analyses revealed that MDA-MB-468 xenografts treated with EC359 + HDACi combination showed decreased proliferation as measured by Ki67 compared to vehicle and monotherapy (Fig. 3d). IHC analyses also revealed that MDA-MB-468 xenografts treated with HDACi have increased STAT3 phosphorylation and treatment with EC359 + HDACi combination attenuated STAT3 phosphorylation (Supplementary Fig. 5). These results suggest that EC359 has potential to enhance the efficacy of HDACi to reduce TNBC xenograft tumor growth.

**EC359 blocks HDACi-mediated LIFR downstream signaling pathways.** We next conducted mechanistic studies using four different TNBC models and three HDACi including vorinostat, panobinostat, and romidepsin. Western blot analysis demonstrated activation of LIFR signaling pathway upon HDACi treatment as evidenced by increased expression of LIFR and activation of its downstream signaling pathways including STAT3 (Fig. 4a, b), AKT, and mTOR (Fig. 4b). Interestingly, pretreatment of EC359 resulted in blockage of LIFR activated downstream signaling in all three HDACi treatments (Fig. 4a, b). Importantly, in STAT3 reporter assays, as expected HDACi treatment stimulated the STAT3-Luc reporter, but pretreatment with EC359 abrogated HDACi induced activation of STAT3 (Fig. 4c, d). Collectively, these findings suggest that HDACi treatment induces expression of LIFR, which in turn contributes to the upregulation of LIFR downstream signaling, however, EC359 is efficacious in reducing HDACi-mediated aberrant LIFR signaling.

**EC359 reduce HDACi-mediated activation of growth-promoting genes and enhance expression of pro-apoptotic genes.** To determine the mechanisms by which EC359 enhanced the potency of HDAC inhibitors, we performed RNA-seq of BT-549 cells treated with vehicle, EC359, HDACi (Vorinostat), and combination (EC359 + HDACi). Comparison of differentially regulated genes (p-adj < 0.01, |log2FC| >1) identified 6341 genes (Fig. 5a, b), which were then subdivided into 6 major clusters by unsupervised clustering (Fig. 5a, Supplementary Fig. 6a). Cluster 1 genes (1157) were synergistically upregulated, while Cluster 3 genes (1501) were synergistically repressed (Supplementary Fig. 6a). Cluster 2 genes (661) are upregulated by HDACi and somewhat suppressed by EC359; while Cluster 4 genes (266) are upregulated by EC359 and somewhat suppressed by HDACi. Cluster 6 genes (1457) are only affected by HDACi and showed

little or no effect with EC359 addition; Cluster 5 genes (1299) are repressed by HDACi and showed no effect with EC359 addition (Supplementary Fig. 6a).

Gene ontology (GO) analyses identified Cluster 1 genes were enriched for signaling pathways associated with the regulation of transcription, cell death, apoptosis, cellular response to stress, and cell differentiation (Fig. 5c); Cluster 3 genes were enriched for pathways of cell cycle, mitosis, cell division, and metabolic process (Fig. 5d). Accordingly, GSEA results showed enrichment of apoptosis and p53 pathways (Fig. 5e), and E2F signaling and Myc pathways (Fig. 5f) for Cluster 1 and Cluster 3 genes, respectively.

We then examined the status of STAT3 target genes using custom-generated STAT3 induced and repressed gene sets that were defined according to published RNA-seq data obtained from TNBC model cell lines[19]. GSEA of these genes revealed that EC359 treatment negatively enriched the STAT3 induced gene set but HDACi treatment showed a trend of positive enrichment of STAT3 induced gene set (Supplementary Fig. 5b). Interestingly, when we compared HDACi monotherapy to combination therapy, we observed a negative enrichment of the STAT3 induced gene set suggesting that the addition of EC359 reverted the HDACi-mediated upregulation of STAT3 activated genes (Supplementary Fig. 5b). On the other hand, both the EC359 and HDACi regulated genes showed a trend for positive enrichment of the STAT3 repressed gene set, however, when compared HDACi monotherapy with combination therapy we observed high positive enrichment of the STAT3 repressed gene set (Supplementary Fig. 5c). Altogether, these results suggest the synergistic activity of EC359 + HDACi combination therapy in part attributed to the ability of EC359 to repress HDACi-mediated aberrant activation of STAT3 target genes and further potentiates the HDACi-mediated upregulation of STAT3 repressed genes.

To validate the RNA-seq findings, we conducted RT-qPCR of selective genes in Cluster 1, 2, and 3. As observed in RNA-Seq, expression of genes involved in apoptosis (Cluster 1) was synergistically enhanced by EC359 + HDACi therapy compared to EC359 or HDACi treatment alone (Fig. 5g). Further, LIF target genes such as ID1, ID2, ID3[20] (genes identified in Cluster 2) are upregulated by HDACi treatment and are reversed by EC359 + HDACi combination therapy (Fig. 5h). Similarly, genes involved in cell cycle (Cluster 3) are further downregulated by EC359 + HDACi therapy compared to EC359 or HDACi treatment alone (Fig. 5i). Altogether, these results suggest that potent activity of EC359 + HDACi combination therapy attributed to the ability of EC359 to enhance HDACi-mediated apoptotic pathways and inhibit genes involved in cell cycle regulation.

**EC359 enhances efficacy of HDACi therapy in PDX derived explant assays.** We used ex vivo culture models of PDX tumors (PDX-derived explants, PDEs) to evaluate the efficacy of EC359 + HDACi treatment. PDEs maintain the native tissue architecture and better recapitulate the heterogeneity of TNBC in

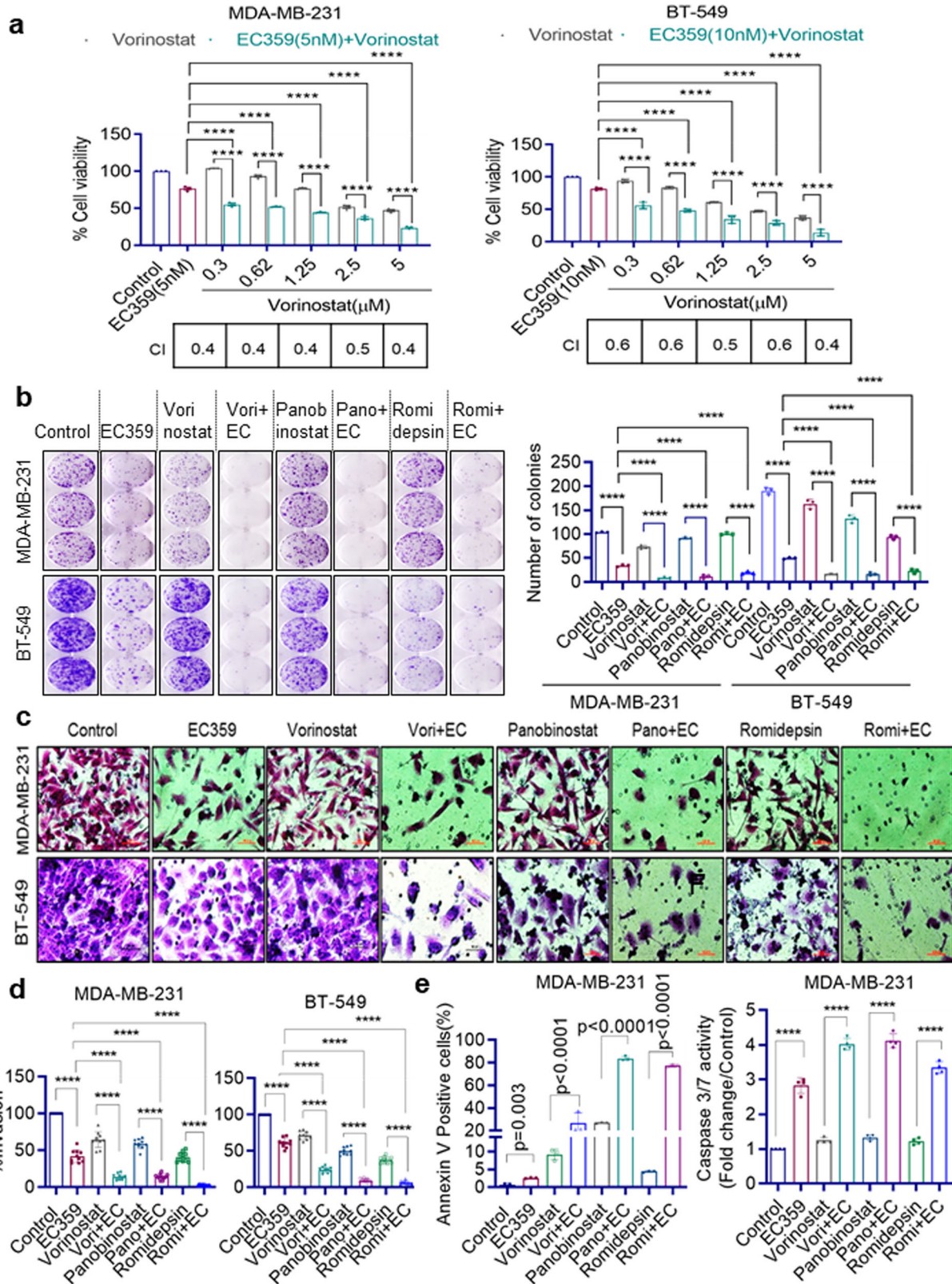

a laboratory setting. We utilized three distinct TNBC PDX tumors and three HDACi (vorinostat, panobinostat, romidepsin) in this assay (Fig. 6a–c). The efficacy of EC359 + HDACi treatment on the proliferation was measured using Ki67 staining. Combination treatment of TNBC PDX tissues revealed that

EC359 enhanced ability of all three HDACi in reducing the proliferation (Ki67 positivity) compared to monotherapy treated tumors (Fig. 6b, c). These results suggest that combination of EC359 + HDACi is more effective in reducing proliferation of TNBC PDEs compared to monotherapy.

**Fig. 2 EC359 synergistically enhanced HDACi ability to reduce cell viability, colony formation, invasion, and to induce apoptosis of TNBC cells.** MDA-MB-231 and BT-549 cells were treated with indicated concentrations of vorinostat (**a**), for 72 h in the presence or absence of EC359 (MDA-MB-231: 5 nM; BT-549: 10 nM) and the cell viability was measured by MTT assay ($n = 3$). Combination Index (CI) values with respect to different concentrations were shown in the bottom of each graph. **b** Effect of EC359 + HDACi combination therapy on the cell survival of TNBC cells was measured using colony formation assays ($n = 3$). Representative images from three independent experiments are shown on the left panel, and quantitation of colonies is presented on the right panel. **c** Effect of EC359 + HDACi combination therapy on cell invasion of MDA-MB-231 and BT-549 cells was determined using matrigel invasion chamber assays. Data are representative of three independent experiments ($n = 3$). Representative images of invaded cells are shown and the number of invaded cells were quantitated (**d**). **e** Effect of EC359 + HDACi therapy on apoptosis was determined using Annexin V-PI staining and Caspase 3/7 activity (Caspase-Glo 3/7® assay) in MDA-MB-231 cells. Data are representative of three independent experiments ($n = 3$). Error bars represent SD. The combination index (CI) of EC359 + HDACi therapy was determined using Chou-Talalay method. $p$-values were calculated using two-way ANOVA. ****$P < 0.0001$.

**EC359 enhances efficacy of HDACi therapy in PDX models.** We next evaluated in vivo efficacy of EC359 and HDACi combination using two TNBC-PDX models. Results showed that EC359 and HDACi (vorinostat) combination therapy is more efficient in reducing the PDX tumor growth compared to EC359 or HDACi monotherapy (Fig. 7a, b). Western blot analysis of tumor lysates showed that PDX mice treated with HDACi have increased expression of LIFR. Interestingly, EC359 and HDACi combination therapy substantially reduced the HDACi-mediated increase in LIFR expression (Fig. 7c). Further, EC359 + HDACi treated tumors showed less proliferation (Ki67 staining) compared to EC359 or HDACi monotherapy (Fig. 7d, e). Collectively, these results suggest that EC359 + HDACi combination therapy is highly effective in reducing the progression of TNBC PDX tumors in vivo.

## Discussion

Epigenetic changes are implicated in the progression of many cancers including TNBC. The FDA has approved HDACi as a class of anticancer agents, for treatment of various cancers. While clinical trials with HDACi in TNBC are ongoing, recent studies suggest that HDACi could have limited efficacy in TNBC due to aberrant activation of LIFR signaling. Here, we tested the hypothesis that a first-in-class LIFR inhibitor, EC359, will have utility in enhancing the potency of HDACi by reducing LIFR oncogenic signaling. The utility of the combination therapy of EC359 with HDACi was tested using multiple TNBC cell lines, cell line-derived xenografts, patient-derived xenograft, and PDX-derived explants. Our findings indicate that EC359 was synergistic with HDACi in multiple TNBC models. Mechanistic studies using multiple TNBC models, CRISPR KO, and RNA-seq, models established the importance of blocking LIFR signaling to enhance the utility of HDACi for treating TNBC.

HDACi have shown promise in treating hematological malignancies. Several HDACi targeting class I, II, and IV HDACs are currently under development for use as anticancer agents[8]. However, their efficacy in treating solid cancers such as TNBC, as monotherapy is limited[5,8]. Preclinical trials using HDACi such as panobinostat, vorinostat, and entinostat have shown that these epigenetic agents exert an anti-proliferative effect on TNBC cells[4,5]. Further, studies showed that HDACi therapeutic outcomes against TNBC improved when they are used in combination with existing chemotherapies, kinase inhibitors, and autophagy inhibitors[5]. Vorinostat is a pan-HDACi that induces apoptosis in several types of hematological and solid tumor cells[5,21]. A recent study showed that HDACi promote activation of LIFR that restrains the efficacy of HDACi in BC[15] and concurrent inhibition of BRD4 or JAK sensitizes TNBC to HDACi[15]. Our study further extended these observations using four distinct HDACi (vorinostat, romidepsin, givinostat, and panobinostat) and provide evidence that HDACi treatment aberrantly activates LIFR signaling. Our earlier studies confirmed the specificity of EC359 using MST, SPR, and LIFR KO TNBC models[16]. Further, EC359 has limited activity in ER+ BC cells, however, it showed potent activity in TNBC models suggesting EC359 effects may be subtype-specific and are independent of steroid receptors such as ER and PR. In this study, we provide evidence that HDACi-mediated activation of STAT3 was abolished in LIFR KO cells and that blocking LIFR signaling using a first-in-class LIFR inhibitor, EC359, enhances HDACi therapy.

The LIFR does not have any intrinsic tyrosine kinase activity, however, LIFR constitutively associates with the JAK-Tyk family of cytoplasmic tyrosine kinases and LIF binding to LIFR complex activates the JAK/STAT pathway[9]. Furthermore, the LIF/LIFR axis can activate multiple signaling pathways including STAT3, MAPK, AKT, and mTOR[11,14] all of which are implicated in TNBC progression. Recent studies suggest the critical role of LIF/LIFR signaling in TNBC progression. TNBC exhibits autocrine stimulation of the LIF-LIFR axis, and overexpression of LIF is associated with poorer relapse-free survival in BC patients[11]. Our results showed that TNBC cells express LIFR co-receptor gp130 and multiple LIFR ligands, thus has the potential to create autocrine/paracrine signaling loops. Accordingly, TNBC exhibited high endogenous STAT3 phosphorylation compared to ER+ BC[16] and EC359 treatment reduced STAT3 phosphorylation in TNBC cells. LIF and LIFR expression occur more prominently in TNBC compared to ER+ BC[16]. Our results using breast tumor tissues are also in agreement with published studies that TNBC tumors have higher level of LIFR. Further, our mechanistic studies demonstrated that EC359 was effective in reducing the aberrant LIFR signaling induced by HDACi treatment in TNBC cells.

LIF/LIFR axis is implicated in tumor growth by modulating several pathways implicated in tumor progression[22] including maintenance of stem cells[17,18]; development of chemoresistance[23,24] and invasion[25]. Tumors upregulate LIF/LIFR signaling via autocrine and paracrine mechanisms[24,26,27]. Several recent studies demonstrated that LIF blockade slow tumor progression, augment the efficacy of chemotherapy[12], and improves therapeutic outcome[13]. Another study using a soluble version of the LIFR as a ligand trap demonstrated that blocking of LIF signaling slows tumor progression in a mouse model of pancreatic cancer[28]. Since TNBC cells express multiple LIFR ligands (LIF, OSM, and CNTF) and co-receptor gp130, aberrant induction of LIFR by HDACi has the potential to contribute to LIFR autocrine signaling loop. LIFR is also reported to function as a metastasis suppressor through the Hippo-YAP pathway[29] and confer a dormancy phenotype in breast cancer cells disseminating to bone[30]. LIFR signaling is complex as multiple ligands activate LIFR including LIF, CNTF, OSM, and CTF1, and LIF/LIFR axis promote TNBC progression[16]. The differences in signaling outcome in different studies may in part arise from differential levels of activation of these pathways, multiple ligands to LIFR, and differences in tumor micro environment (TME)[31,32]. Recently published studies indeed support oncogenic

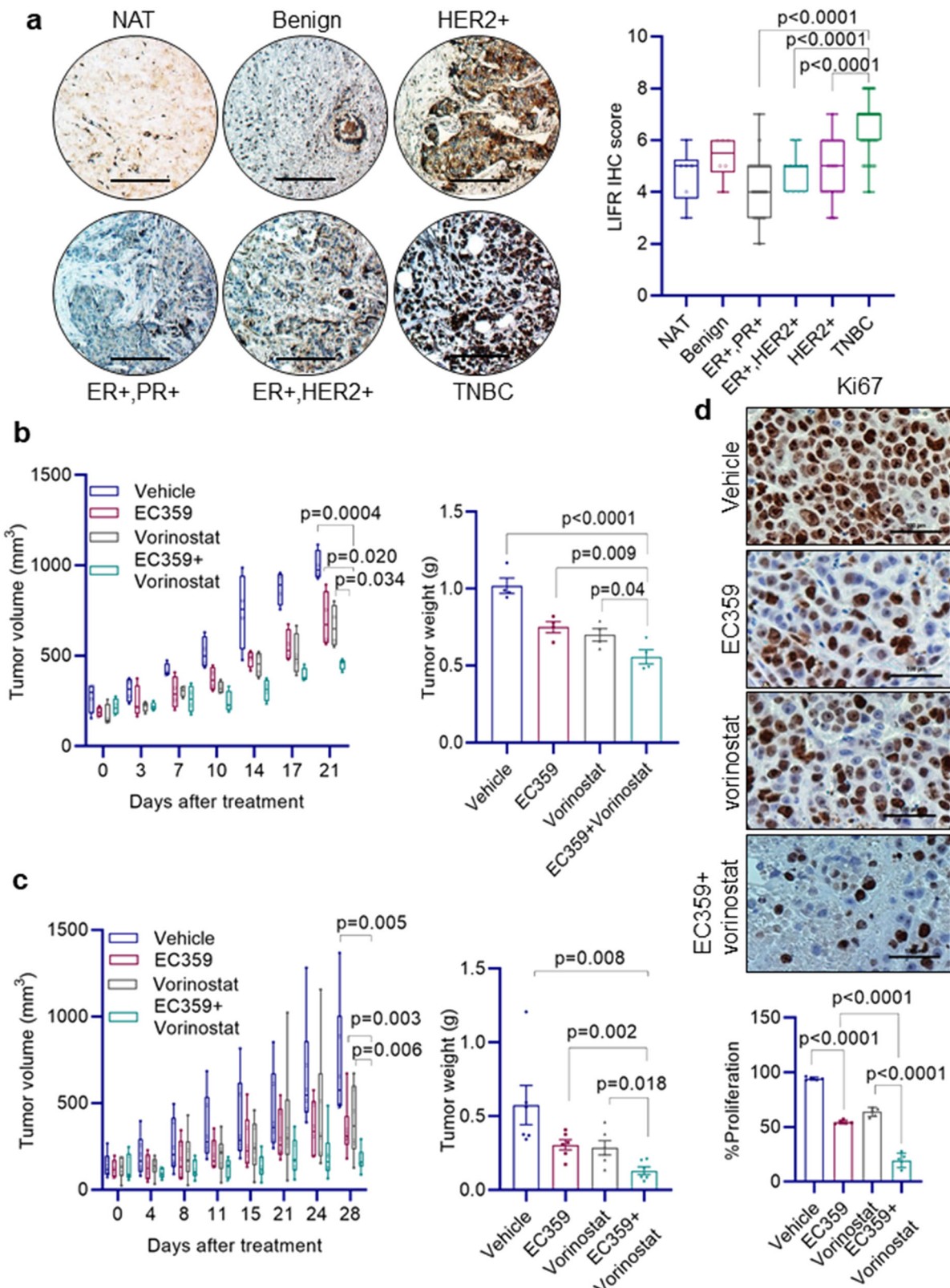

role of LIF/LIFR signaling in cancer progression[16,33,34]. Our results from RNA-seq studies also suggest that the beneficial effect of the EC359 + HDACi involves regulation of multiple genes that involved in several pathways including apoptosis, metabolism, and cell cycle.

Recent studies suggested LIF/LIFR axis as a promising clinical target for cancer therapy[12,13]. Considering the importance of LIF/

LIFR pathway, Northern Biologics Company recently developed a humanized Anti-LIF antibody (MSC-1) that blocks LIF signaling, and its utility is being tested in phase I clinical trial mode to determine its safety and tolerability (ClinicalTrials.gov, NCT03490669). However, lack of small molecule inhibitors that block LIF/LIFR signaling represents a major knowledge gap. Further developing a small molecule inhibitor targeting LIF/LIFR

**Fig. 3 LIFR is highly expressed in TNBC and EC359 + HDACi combination therapy reduced TNBC xenograft tumor growth. a** Breast tissue microarray consisting of normal adjacent tissue (NAT, $n = 6$), benign ($n = 6$), ER$^+$PR$^+$ ($n = 50$), ER + HER2+ ($n = 11$), HER2+ ($n = 30$), and TNBC ($n = 61$) samples were subjected to immunohistochemical staining using LIFR antibody and the intensity and positivity of LIFR staining was quantitated. Scale bar represents 100 μm. **b** MDA-MB-231 ($n = 4$) and **c** MDA-MB-468 ($n = 6$) xenografts in SCID mice were treated with vehicle or EC359 (5 mg/kg/day) or vorinostat (100 mg/kg/day) or in combination. Tumor volume and weight of vehicle and treated tumors were measured. **d** the proliferation of MDA-MB-468 xenograft tumors was determined using Ki67 immunostaining. Representative Ki67 staining from each treatment condition is shown in the upper panel and quantification plot is shown in the lower panel. In **a**, p-value was calculated using one-way ANOVA. Error bars represent SD. In **b**, **c**, and **d**, p-values were calculated using t test and two-way ANOVA.

is very cost effective. Recently, we developed a first-in-class inhibitor of LIFR, EC359. Our results using multiple in vitro, ex vivo, and in vivo TNBC models and PDX models demonstrated the utility of EC359 in enhancing the efficacy of HDACi for treating TNBC.

TNBC are very heterogeneous tumors and often exhibits autocrine stimulation of LIFR axis by alternating the expression of ligands. Endoplasmic reticulum stress and hypoxia are few of the hallmarks of TNBC[35]. Hypoxia induces LIF expression in human cancer cells[36]. LIF promotes proliferation and metastasis of BC cells, and overexpression of LIF is commonly associated with poorer relapse-free survival in BC patients[11]. Further, increased expression of alternative LIFR ligands such as OSM, CNTF, and CTF1 was reported in TNBC. EC359 was shown to inhibit the LIFR activation by all LIFR ligands including LIF, OSM, CNTF, and CTF1[16]. Since LIFR activates multiple downstream signaling pathways including STAT3, AKT, and mTOR, blockage of STAT3 alone may not enhance HDACi efficacy. Similarly, use of BRD4 inhibitors alone may not increase the efficacy of HDACi as TNBC utilize multiple ways to activate LIF-LIFR signaling. In support of this, our results suggest that EC359 is highly effective in enhancing therapeutic utility of HDACi for treating TNBC.

In summation, our results using multiple TNBC models suggest that aberrant activation of LIFR occurred upon HDACi treatment and further demonstrated the utility of LIFR inhibitor EC359 in enhancing HDACi therapy. Since HDACi and LIFR inhibitors are currently in clinical development, the findings from this study suggest that a combination of the two agents may provide additional clinical benefit in the treatment of TNBC.

## Methods

**Cell culture and reagents.** Human TNBC cells (MDA-MB-231, BT-549, MDA-MB-468, HCC1806, and HCC70) were purchased from the American Type Culture Collection (ATCC, Manassas, VA) and cultured as per ATCC guidelines. All the model cells utilized were free of mycoplasma contamination. STR DNA profiling was used to confirm cell identity. Antibodies for GAPDH, p-S6, S6, p-Akt (S473), Akt, p-mTOR (S2448), mTOR, p-STAT3 (Y705), and STAT3 were purchased from Cell Signaling Technology (Danvers, MA). LIF and LIFR (LIFRα, CD118) antibodies were purchased from Santa Cruz Biotechnology (Dallas, TX). β-actin antibody was obtained from Sigma-Aldrich (St. Louis, MO). The Ki67 antibody was purchased from Abcam (Cambridge, MA). LIFR knockout (KO) model cells and synthesis of EC359 were done using protocol in our earlier publication[16]. Vorinostat, panobinostat, romidepsin, and givinostat were purchased from MedChemExpress (Monmouth Junction, NJ).

**Cell viability, colony formation, invasion, and apoptosis assays.** The effects of vehicle (DMSO), EC359, and HDACi alone or in combination on cell viability were measured using the MTT cell viability assay[16]. HDACi and EC359 concentrations used in the assays were based on our earlier studies[16], dose–response curves using TNBC models, and published studies[15]. The combination index (CI) of EC359 + HDACi therapy was determined using Chou-Talalay method[37]. Apoptosis was measured using Annexin V-PI staining (BioLegend, San Diego, CA) and Caspase-Glo® 3/7 assays (Promega, Madison, WI) according to manufacturer's protocol. For colony formation assays, TNBC model cells (500 cells/well) were seeded in triplicate in 6-well plates, treated with indicated drugs for 5 days, and allowed to grow for 14 days. The cells were fixed in ice-cold methanol and stained with 0.5% crystal violet solution. Colonies that contain ≥50 cells were counted and used in the analysis. The effect of combination therapy on cell invasion was studied using the Corning® BioCoat™ Growth Factor Reduced Matrigel® Invasion Chamber assay

(Corning, Corning, NY). MDA-MB-231 and BT-549 cells were treated with vehicle or EC359 or HDACi or combination for 22 h and invaded cells were determined and quantitated using the manufacturer's protocol.

**Western blotting and RT-qPCR.** Whole-cell lysates were prepared using RIPA buffer as previously described[16]. Total proteins were mixed with SDS sample buffer and subjected to SDS-PAGE. All primary antibodies for western blotting were done using 1:1000 dilution. Secondary antibodies for western blotting were diluted 1:1000 for anti-mouse antibodies and 1:2000 for anti-rabbit antibodies. Blots were developed using the ECL kit (Thermo Fisher Scientific, Waltham, MA). RT-qPCR was performed using SuperScript III First Strand kit (Invitrogen, Carlsbad, CA) and by using SYBR Green on an Illumina Real-Time PCR system. Primer sequences of the genes used were included in Supplementary Table 1.

**Luciferase reporter assays.** Generation of MDA-MB-231 and BT-549 cells stably expressing a STAT3-luciferase reporter was earlier described[16]. For reporter assays, cells were serum starved for 24 h, treated with EC359 or HDACi or combination therapy for 24 h, and reporter activity was measured. Cells were lysed in passive lysis buffer, and the reporter activity was measured using the dual-luciferase reporter assay system (Promega, Madison, WI).

**CSCs cell viability and mammosphere formation assays.** Cancer stem cells (CSCs) from BT-549 cells were sorted using the ALDEFLUOR kit (STEMCELL Technologies, Cambridge, MA) and flow cytometry. CSCs were cultured in MammoCult medium as per manufacturer's instructions. The effect of EC359 + HDACi combination therapy on the viability of CSCs was measured using Cell Titer-Glo assays (Promega, Madison, WI, USA). For CSC mammosphere assays, single-cell suspensions of CSCs were seeded in 24-well ultra-low attachment plates (100 cells/well) in triplicate and treated with vehicle or EC359 or vorinostat or EC359 + vorinostat for 7 days, and the newly formed spheres were counted.

**RNA-seq and differential expression analysis.** The effect of EC359 + HDACi combination therapy on global transcriptome was determined by RNA-sequencing using established protocol[38]. Total RNA from BT-549 cells treated with vehicle, EC359, vorinostat, and EC359 + vorinostat was prepared using RNeasy mini kit (Qiagen, Valencia, CA). Illumina TruSeq RNA Sample preparation and sequencing were performed at Greehey Children's Cancer Research Institute Genome Sequencing Facility (UTHSA) using standard illumina HiSeq 3000 sequencing protocol. The raw reads were aligned to the reference human genome (UCSC hg19) with TopHat2[39]. Genes were annotated (using NCBI RefSeq) and quantified by HTSeq[40], and DESeq[41] was used to identify differentially expressed genes and genes with fold change >1 and multiple-test adjusted p value <0.01 were used for interpreting the biological pathways. Gene Set Enrichment Analysis was performed using stand-alone distribution (http://www.gsea-msigdb.org/gsea/index.jsp)[42]. STAT3 induced gene set and a STAT3 repressed gene set were custom-generated using published TNBC model RNA-seq data[19]. For GSEA, due to the size limitation of gene set (<500), we choose the top 500 STAT3 induced genes and 341 STAT3 repressed genes ($p < 0.01$, $log2FC < -0.5$). These 2 gene sets are close to our RNA-seq data (both cell lines are TNBC mesenchymal cells). RNA-seq data have been deposited in the GEO database under a GEO accession number GSE163249.

**Animal studies.** All animal experiments were performed using UTHSA IACUC approved protocol. For xenograft tumor assays, $2 \times 10^6$ MDA-MB-231 ($n = 4$) or MDA-MB-468 ($n = 6$) cells were mixed with an equal volume of matrigel and implanted in the mammary fat pads of 6-week-old female athymic nude mice[16]. Once tumors reached measurable size, mice were divided into control and treatment groups. Mice bearing TNBC PDX tumors were purchased from Jackson laboratory (TM00089; TM00096; TM00098) and TNBC PDX line UTPDX0001 establishment was earlier described[16]. When tumors reached ~750 mm$^3$ they were dissected into 2 mm$^3$ pieces and implanted into the flanks of 6-week-old female SCID mice. When the tumor volume reached ~150 mm$^3$, mice were randomized for treatment. The control group received vehicle and the treatment groups received EC359, or vorinostat or EC359 + vorinostat. EC359 and vorinostat doses were chosen based on earlier published studies[15,16]. The mice were monitored daily for adverse toxic effects and tumor volume was measured every 3-4 days using

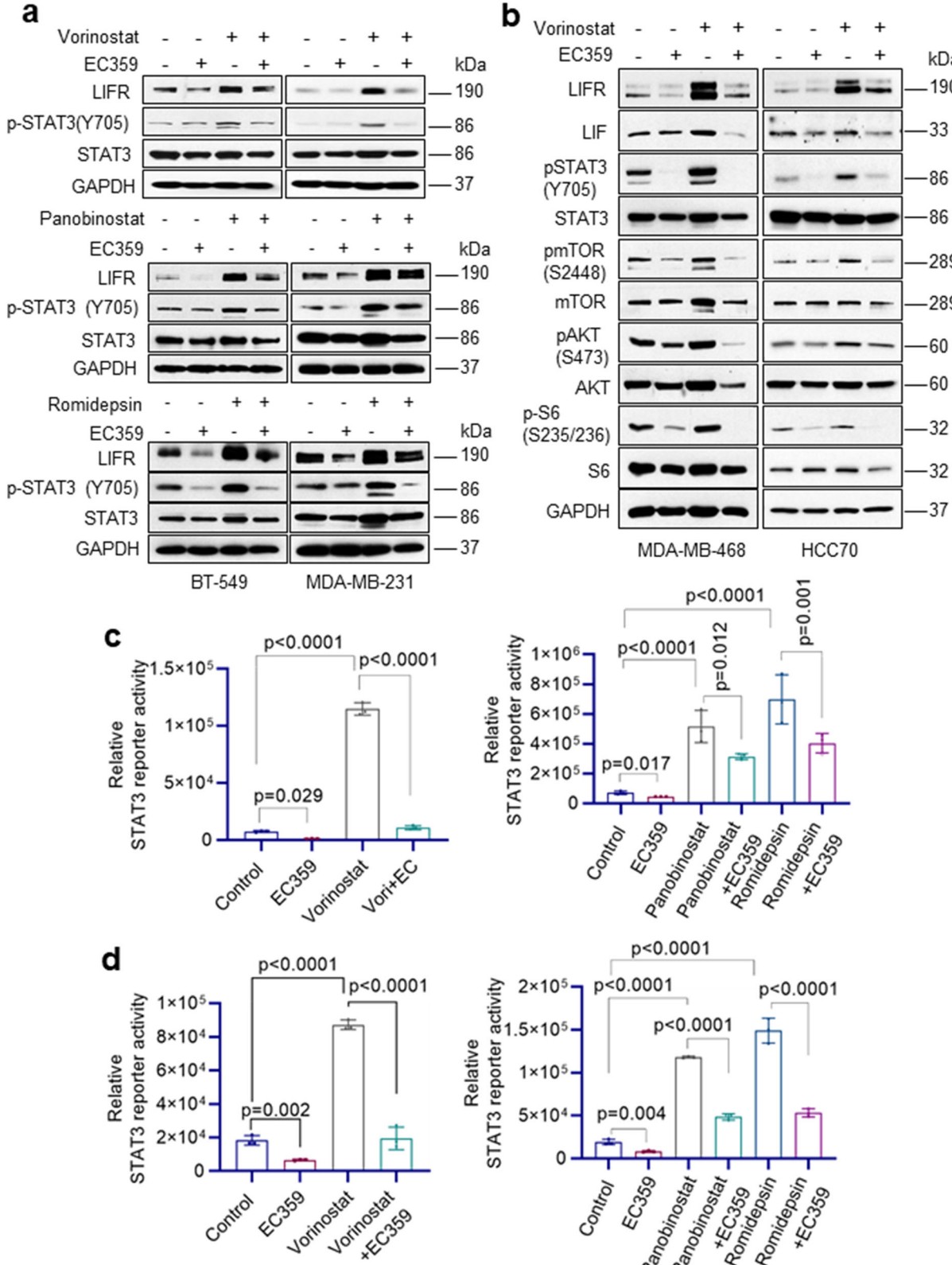

**Fig. 4 EC359 inhibits HDACi induced LIFR signaling in TNBC cells. a** BT-549 and MDA-MB-231 cells were treated with EC359 (100 nM) or HDACi (vorinostat (10 μM) or panobinostat (500 nM) or romidepsin (100 nM)) alone or in combination for 10 h. Effect of EC359 on HDACi induced LIFR downstream signaling was measured using western blotting. **b** MDA-MB-468 and HCC70 cells were treated with EC359 (100 nM) or vorinostat (10 μM) alone or in combination. Effect of EC359 on vorinostat-induced LIFR downstream signaling was measured using western blotting. Effect of EC359 on HDACi induced STAT3-Luc activity in BT-549 (**c**) and MDA-MB-231 (**d**) cells was determined using reporter assays. Data are representative of three independent experiments ($n = 3$). Error bars represent SD. In **c**, **d**, $p$-values were calculated using two-way ANOVA.

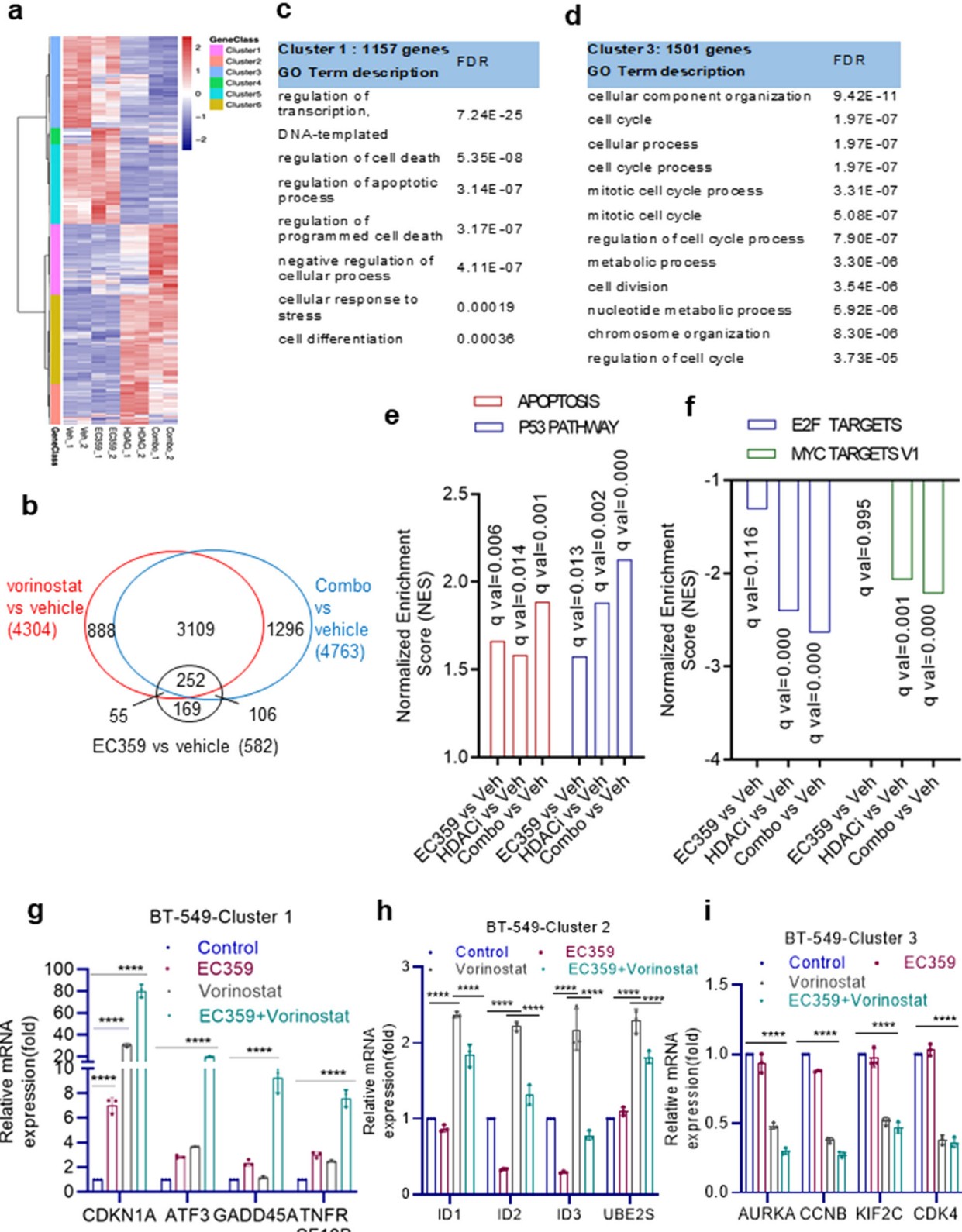

**Fig. 5 Global transcriptomic analyses of EC359 + HDACi combination therapy in TNBC cells.** BT-549 cells were treated with EC359, vorinostat, or EC359 + vorinostat for 24 h, and RNA-seq analysis was conducted. **a** Heat map depicts the clustering of all genes with RPKM >1 (p-adj<0.01, |log2FC| >1). **b** Venn diagram comparing differentially regulated genes (p-adj < 0.01, |log2FC| >1) identified from RNA-seq data. **c**, **d** representative pathways identified using GO term description in Cluster 1 and 3 are shown. **e**, **f** Top pathways identified in Cluster 1 and 3 using HALMARK GSEA are shown. **g–i** RT-qPCR validation of selective genes from Cluster 1, 2, and 3 is shown. Error bars represent SD. ****$P < 0.0001$.

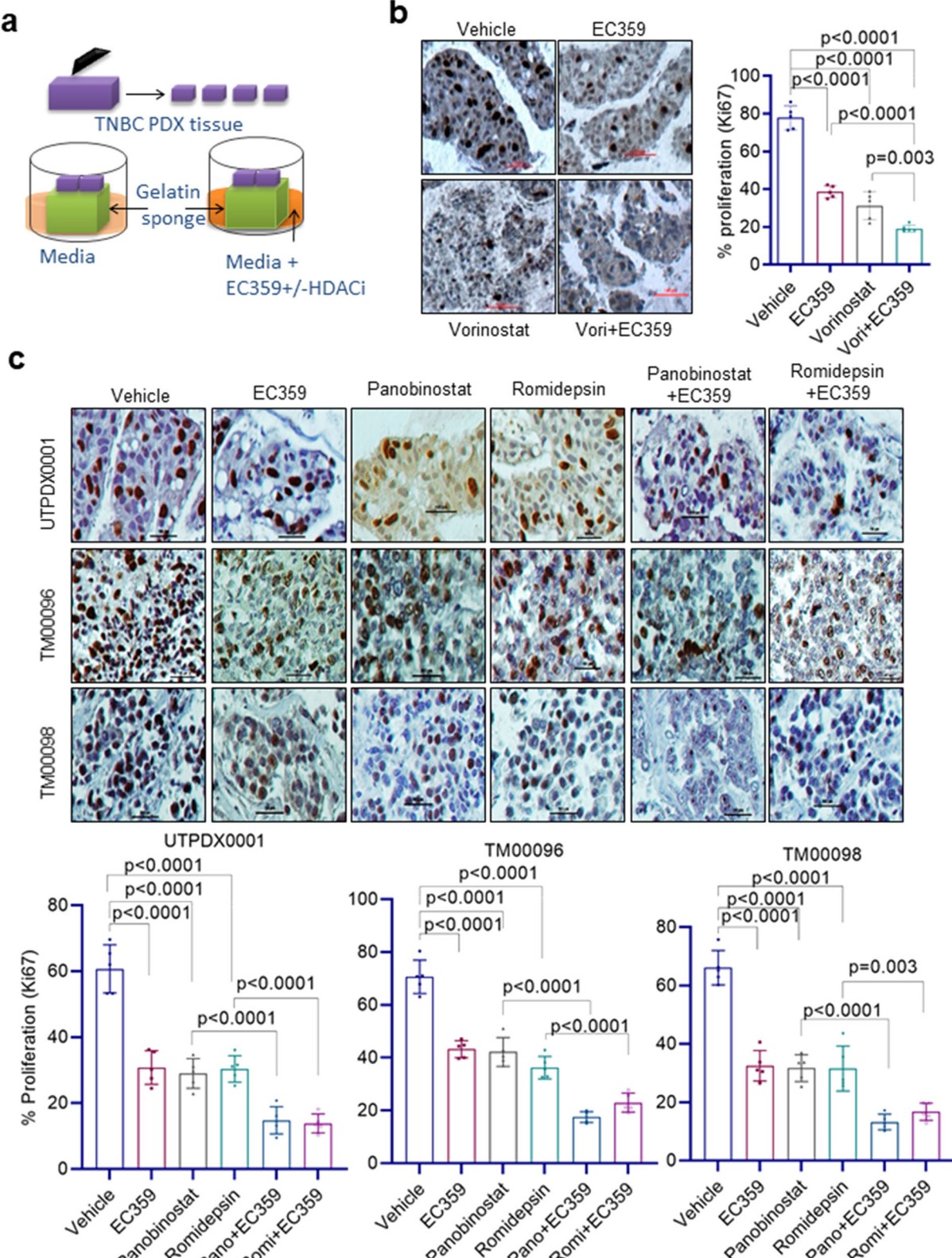

**Fig. 6 EC359 enhance the ability of HDACi to decrease the proliferation of patient-derived xenograft explants (PDE) ex vivo. a** Schematic representation of ex vivo culture model. **b** TNBC PDX explants (UTPDX0001) were treated with EC359 (500 nM), vorinostat (10 μM), or combination for 72 h and the proliferation was determined using Ki67 immunostaining. Data are representative of three independent experiments ($n = 3$). **c** TNBC explants from three different PDX tumors (UTPDX0001, TM00096; TM00098) were treated with EC359 (500 nM), panobinostat (100 nM), romidepsin (50 nM) or combination for 72 h and the proliferation was determined using Ki67 immunostaining. Representative Ki67 staining from each treatment condition is shown. Data are representative of three independent experiments ($n = 3$). The number of Ki67-positive cells from five different images were counted and plotted as histogram. Error bars represent SD. In **b**, and **c**, p-values were calculated using one-way ANOVA.

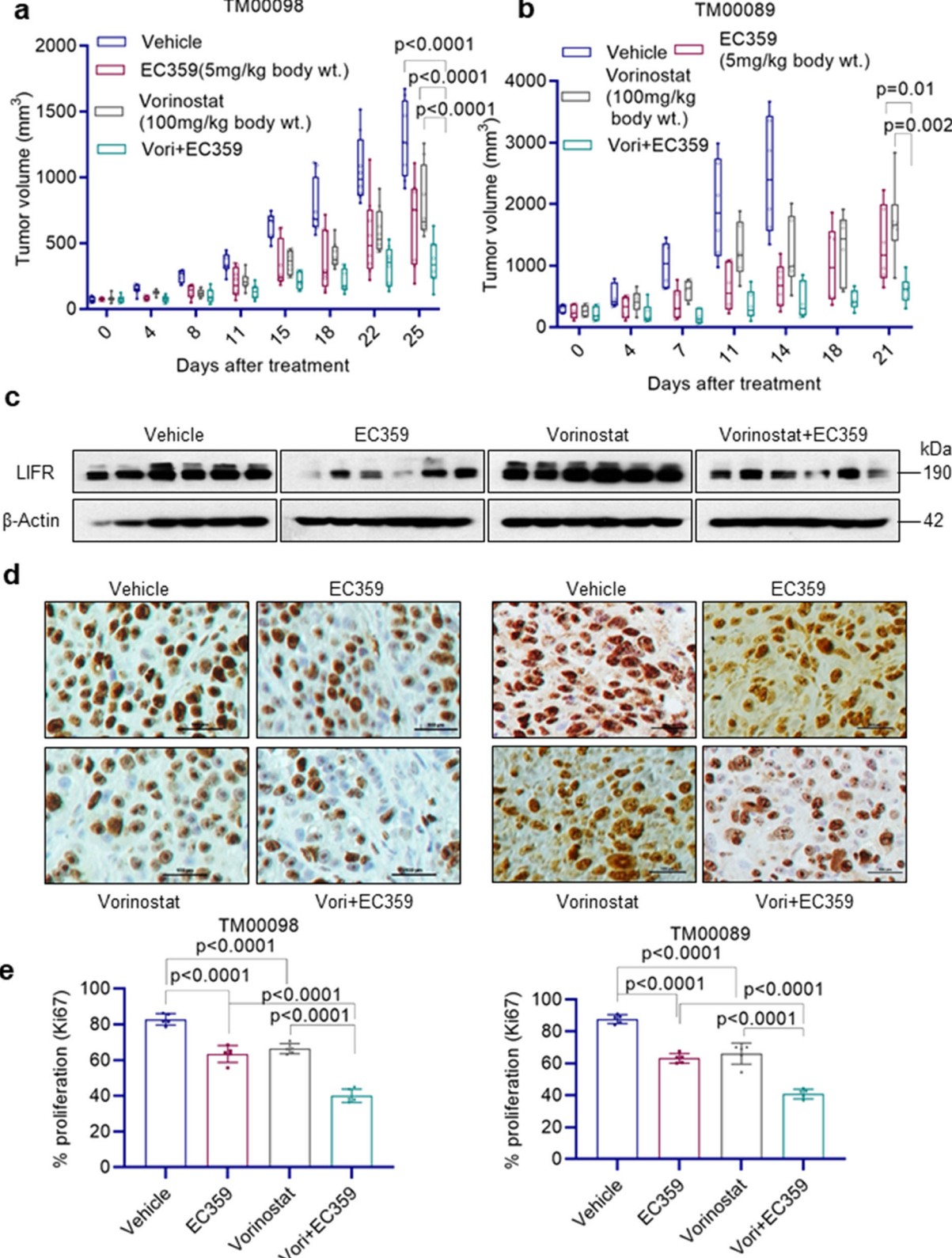

**Fig. 7 EC359 enhanced HDACi ability to reduce tumor growth in PDX models of TNBC. a**, **b** Female SCID mice were implanted with 2 mm³ pieces of PDX tumors TM00098 (**a**) TM00089 (**b**) into the flanks and randomized for treatment when the tumor volume reached ~150 mm³. TNBC PDX tumors ($n = 6$–$8$) were treated with vehicle or EC359 or vorinostat or combination. Tumor volumes are shown in the graph. Control mice in panel **b** were euthanized on day 14 because of higher tumor volume. **c** Total protein lysates from vehicle and treated TM00089 PDX tumors were analyzed for levels of LIFR using western blotting. **d**, **e** Ki67 expression as a marker of proliferation was analyzed by IHC and quantitated. Error bars represent SD. In **a**, p-values were calculated using two-way ANOVA. In **b**, p-values were calculated using multiple unpaired t test. In **e**, p-values were calculated using one-way ANOVA.

calipers. At the end of each experiment, the mice were euthanized, and the tumors were removed, weighed, and processed for histological studies and protein analysis. IHC analysis was performed using published protocol[16]. Primary antibodies were incubated overnight with Ki67 (1:100) or Phospho STAT3 (Y705) antibody (1:50) and subsequent secondary antibody incubation for 30 min at room temperature. Percentage of Ki67-positive proliferating cells was calculated in five randomly selected microscopic fields.

**PDX-derived explant (PDE) studies.** Excised PDX tumor samples (UTPDX0001; TM00096; TM00098) were processed, and cultured ex vivo as previously described[43,44]. Briefly, PDX tumor samples were dissected into $2 \, mm^3$, and then incubated on gelatin sponges for 24 h in culture medium containing 10% FBS, followed by treatment with either vehicle or HDACi or EC359 or in combination for 72 h. Treated tumor tissues were then fixed in 10% formalin at 4 °C overnight and then processed into paraffin-embedded blocks.

**Tissue microarray (TMA) and immunohistochemistry analysis.** BC tissue microarrays (TMA cat# BR1503g and BR487c) were purchased from US Biomax, Inc. (Rockville, MD). TMAs were probed using LIFR antibody. IHC analysis was conducted[45] and immunoreactivity was visualized using DAB substrate and counterstained with hematoxylin (Vector Laboratories Inc., Burlingame, CA). Tissue arrays were scored using Allred Scoring system[46]. Briefly, the LIFR staining intensity was scored on a scale between zero and three and the proportion of positive stained cells was rated as one between 0 and 1%, two between 1 and 10%, three between 10 and 33%, four between 33 and 66%, and five between 66 and 100%. Control rabbit IgG staining was used as a negative control. The sections were scored by two independent evaluators blinded to the patient's clinical status.

**Statistics and reproducibility.** Statistical differences between groups were analyzed with either a $t$-test or ANOVA as appropriate using GraphPad Prism 9 software. A value of $p < 0.05$ was considered as statistically significant. The combination index (CI) was calculated using the Chou-Talalay method[37]. For animal studies, sample size of tumors/treatment was derived using effect information from previous studies and calculations were based on a model of unpaired data power = 0.8; $p < 0.05$. All in vitro assays were performed in biological replicates in technical triplicate.

**Reporting summary.** Further information on research design is available in the Nature Research Reporting Summary linked to this article.

## Data availability

The data supporting the findings of this study are available from the corresponding authors upon reasonable request. Source data for the graphs and charts are provided in Supplementary Data 1. The RNA-seq data have been deposited in the GEO database (accession number GSE163249). Unedited western blot images are available in the Supplementary Information (Supplementary Figs. 7–17).

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

## Acknowledgements

We thank Perry Jessica (Ob/Gyn UT Health San Antonio) for proofreading the manuscript. This study was supported by the DOD BCRP grant W81XWH-18-1-0016 (R.K.V.; K.J.N.); and DOD BCRP grant W81XWH-18-1-0021(Z.X.; R.K.V.); NCI Cancer Center Support Grant P30CA054174-17; NIH CTSA UL1TR002645; Voelcker fund young investigator award (G.R.S.) CPRIT Pre-doctoral Fellowship (K.A.A.), UT Foundation/ Shelby Tengg Foundation award 166120/44090 (S.V.), Elsa U. Pardee Foundation grant 166675-44096 (S.V), NIH grant 1R01CA179120-01 (R.K.V.; M.R.), VA Grant I01BX004545 (R.K.V) and CPRIT RP160732 (Y.C.). GCCRI Genome Sequencing Facility is supported by NIH Shared Instrument grant 1S10OD021805-01.

## Author contributions

Conception and design: S.V., G.R.S., Z.X., C.Y., M.R., A.J.B., V.G.K., R.R.T., H.B.N., K.J.N., G.R., and R.K.V. Development of methodology: M.L., S.V., B.S., G.A., U.P.P., X.L., H.Y., Y.L., J.L., K.A.A., W.T., and Z.L. Acquisition of data: M.L., S.V., B.S., S.K., U.P.P., Z.Y., X.L., H.Y., Y.L., J.L., K.A.A., W.T., Z.L., and B.E. Analysis and interpretation of data: S.V., G.R.S., C.Y., Z.Y., and J.L. Writing, review, and/or revision of the manuscript: M.L., S.V., G.R.S., K.A.A., G.R., and R.K.V. Administrative, technical, or material support: A.J.B., V.G.K., R.R.T., H.B.N., G.R., K.J.N., and R.K.V. Study supervision: S.V., H.B.N., and R.K.V.

## Competing interests

The authors declare the following competing interests: Authors including Bindu Santhamma, Swapna Konda, Gulzar Ahmed, Hareesh B. Nair, and Klaus J. Nickisch are current employees of Evestra Inc. All other remaining authors declare no competing interests.
