## [Peer Review File · Communications Biology]

Reviewers' comments:

Reviewer #1 (Remarks to the Author):

This study tested the hypothesis that inhibition of LIFR function is synergistic with the inhibition of HDAC when used to treat triple negative breast cancer (TNBC). Utilizing multiple in vitro, ex vivo, and in vivo TNBC models, the authors made two major observations: 1) HDAC inhibition in TNBC cells leads to the increase of LIFR expression and overactivation of the downstream signaling elements such as STAT3, mTOR, and AKT, thus enhances the aggressive potential of TNBC cells (proliferation, invasion, stemness). 2) The LIFR inhibitor EC359, which was developed by the same research group, synergistically enhanced the efficacy of HDAC inhibitors in suppressing TNBC in vitro and in vivo. Based on these observations, the authors concluded that Overall, the experimental design of this study is comprehensive by employing multiple cell lines, testing multiple inhibitors, investigating multiple signaling events and cell behaviors, and utilizing multiple in vitro and in vivo models. Data are of high quality with proper controls included. The major conclusion that EC359 enhances the efficacy of HDACi when applied to TNBC cells and the combined therapy deserves follow-up clinical studies of TNBC treatment is well justified. Therefore, I think this study will be of interest to readers interested in TNBC treatment and merits consideration of publication. Although my overall impression is positive, I also think some mechanistic evidence is weak and needs to be reinforced by additional experiments and/or discussion.

Major points:

1. HDACi treatment increased the transcription and expression of LIFR in cultured cells or in tissues (Figs. 1, 4, and 7). Are these LIFR α or LIFR β ? Why are there two bands recognized by LIFR antibody? Mouse LIFR exists in both the membrane-bound full-length receptor or the soluble form resulted from alternative splicing that lacks the transmembrane domain and cytoplasmic domain. Does human LIFR behave the same? Protein size markers are not provided.

2. How were these upregulated LIFR activated? I didn't see any LIFR ligands were added to cultured cells. Were these upregulated LIFR constitutively activated? Considering that LIFR could bind and be activated by LIF, OSM, CNTF, and CT when forming receptor complex with different co-receptors, such as gp130 and other specific receptor subunit related to IL-6 signaling.

3. Considering the critical role of gp130, i.e. forming heterodimer with LIFR to enable the LIF signaling, it is important to determine how gp130 expression is changed by HDACi and whether gp130 contributes together with LIFR.

Minor points:

1. Fig. 3D is not mentioned in the figure legend.

Reviewer #2 (Remarks to the Author):

In this manuscript, authors identified that Histone deacetylase inhibitors (HDACi) and LIFR inhibitor induces synergistic cell death in TNBC. Authors observed that both targeted knockdown of LIFR with CRISPR or treatment with EC359 significantly enhanced the efficacy of four different HDACi in reducing cell viability, cell survival, and enhancing apoptosis compared to monotherapy in TNBC cells. Mechanistically, RNA-seq studies demonstrated oncogenic/survival signaling pathways activated by HDACi were attenuated by LIFR inhibition. Importantly, combination therapy potently inhibited the growth of TNBC patient derived explants, TNBC xenografts and patient-derived xenografts in vivo. The manuscript is well written and well conceptualized. However, following concerns needs to be addressed.

- In the histogram represented in figure 1F, comparison for statistics between control LIFR-KO cells and LIFR-KO cells treated with HDACis, for better understanding the effect of this combination. Same would be helpful for figure 2A.

- Figure 2B, from the colony images, EC359 shows significant reduction in the colony formation. It is a suggestion for the data clarity that SS may be better compared with the EC359 Vs the

HDACi+EC combination, rather than with the controls. Similarly, interpretation of the same order for figure 2D would be more clear.

- If data available, It is suggested to include Phospho-STAT3 IHC staining in figure 3D or 6C to make sure that the STAT3 signaling mechanism are attenuated upon LIFR inhibition in-vivo.
- Authors mentioned about the methodology for Mammosphere assay. However, no such data was represented in the manuscript. Please clarify.
- Abbreviation of LIFR in the abstract would help the readers.

Reviewer #3 (Remarks to the Author):

1. This is a very impressive study characterizing anti-TNBC action of HDACi in combination with EC359. The results of cell-based studies are very convincing and clearly point at synergistic effect of EC359 with HDACi. Evaluation in xenograft with MDC-MB-468 and MDA-MB-231 cells as well PDX animal models strengthen the benefit of the combination. Gene expression and signaling analysis enhance understanding of the mechanistic events underlying the action of the compounds.

2. The authors already characterized the activity of EC359 against GR and ER in their previous publication. This compound, however, is incredibly similar to other steroids acting as ligands/agonists/antagonists/blockers for a variety of steroid hormone receptors. One cannot exclude that EC359 is capable of both binding to LIFR and modulation of steroid hormone receptor(s). It would very useful to confirm that the results observed are not a result of steroid hormone receptor modulation and LIFR/LIF interaction blocking using a wider panel of steroid hormone receptors. The authors already tried to do it in this manuscript and in the paper where EC359 was reported. Unfortunately, STAT3, AKT, and mTOR seem to be too general to attribute to LIFR signaling only and also have well-known crosstalk with multiple steroid hormone receptors. The authors already have LIFR-KO cells that can be treated with a combination of EC359 and HDACi. If LIFR is indeed the only target of EC359, the effect of the combination should be indistinguishable from that of HDACi only.

3. The concentrations of HDACi are very high. Why? This was not explained.

4. Error bars should be shown as SD, not SEM. SD is a measure of variability, which is the most relevant metric in these experiments.

5. The authors keep using "therapies" when describing EC359 and HDACi and "therapeutic efficacy" when describing experiments with cells in culture. This use of "therapies" or "therapeutic efficacy" is incorrect. EC359 or HDACi are not a therapy in this context and cells are not animals or patients. The use of "efficacy" is generally incorrect when used for experiments in cells in culture. The authors measure biological effect in cells in culture as a function of HDACi concentration, which is potency. In those cases, the words "efficacy" should be replaced with "potency".

Response to reviewer comments

We greatly appreciate the **overall positive comments of reviewers and editor** that “the experimental design of this study is comprehensive by employing multiple cell lines, testing multiple inhibitors, investigating multiple signaling events and cell behaviors, and utilizing multiple in vitro and in vivo models”, “Data are of high quality with proper controls included”, “The major conclusion that EC359 enhances the efficacy of HDACi when applied to TNBC cells and the combined therapy deserves follow-up clinical studies of TNBC treatment is well justified”, “ this study will be of interest to readers interested in TNBC treatment and merits consideration of publication”, “The manuscript is well written and well conceptualized”, “This is a very impressive study characterizing anti-TNBC action of HDACi in combination with EC359.” We have now conducted additional experiments as suggested by reviewers and modified the manuscript based on the editor and reviewer’s suggestions as detailed below. All the changes in the text are shown using blue color font.

Reviewer #1:

Major points:

1. HDACi treatment increased the transcription and expression of LIFR in cultured cells or in tissues (Figs. 1, 4, and 7). Are these LIFR α or LIFR β ? Why are there two bands recognized by LIFR antibody? Mouse LIFR exists in both the membrane-bound full-length receptor or the soluble form resulted from alternative splicing that lacks the transmembrane domain and cytoplasmic domain. Does human LIFR behave the same? Protein size markers are not provided.

Response: The RT-qPCR and Western blots showed the expression of LIFR (referred to as LIFR α , LIF Receptor Subunit Alpha, CD118). LIFR α is a glycosylated protein and it run as doublets of 190 and 170 kDa on western blots and differences in sizes are due to level of glycosylation (ref PMID: 10858440). Further, the primers or antibody used in this study are specific to LIFR alpha and they do not recognize or cross-react with gp130 (130 kDa protein), an high-affinity converter subunit of LIFR alpha. Low amounts of soluble LIFR reported in both mouse and humans. Its amount increases (30-fold) in the late pregnancy. Alternative splicing and introduction of stop codon before transmembrane contributes to generation of soluble LIFR. It is suspected to play a role in neutralizing LIF and other LIFR ligands during pregnancy. The Kd of human LIF to soluble LIFR is 7.8nM while to membrane LIFR is 87pM. Soluble LIFR unable to generate the signals in the cells as cytoplasmic domain of LIFR is required for signaling. In epithelial cancers such as TNBC, increase of membrane LIFR predominantly occur. We have now included protein size markers for Western blots.

2. How were these upregulated LIFR activated? I didn’t see any LIFR ligands were added to cultured cells. Were these upregulated LIFR constitutively activated? Considering that LIFR could bind and be activated by LIF, OSM, CNTF, and CT when forming receptor complex with different co-receptors, such as gp130 and other specific receptor subunit related to IL-6 signaling.

Response: The upregulated LIFR in TNBC cells is constitutively activated as TNBC cells express LIFR ligands and exhibits autocrine LIFR signaling. Our Western blot results confirm that TNBC models indeed express LIF (Fig. 1d, 4b). We have now included RT-qPCR data showing that TNBC cells do express gp130 and LIFR ligands OSM and CNTF (Extended data Fig. 1b).

3. Considering the critical role of gp130, i.e. forming heterodimer with LIFR to enable the LIF signaling, it is important to determine how gp130 expression is changed by HDACi and whether gp130 contributes together with LIFR.

Response: We examined the expression of gp130 in two different TNBC models. Results showed that these model express gp130 and its expression levels are not altered by HDACi treatment (Extended data Fig. 1c).

Minor points:

1. Fig. 3D is not mentioned in the figure legend.

Response: We have now cited Fig 3D.

Reviewer #2 (Remarks to the Author):

1. In the histogram represented in figure 1F, comparison for statistics between control LIFR-KO cells and LIFR-KO cells treated with HDACis, for better understanding the effect of this combination. Same would be helpful for figure 2A.

Response: Thanks for this suggestion. We have now included this comparison.

2. Figure 2B, from the colony images, EC359 shows significant reduction in the colony formation. It is a suggestion for the data clarity that SS may be better compared with the EC359 Vs the HDACi+EC combination, rather than with the controls. Similarly, interpretation of the same order for figure 2D would be more clear.

Response: Thanks for this suggestion. We have now included this comparison.

3. If data available, It is suggested to include Phospho-STAT3 IHC staining in figure 3D or 6C to make sure that the STAT3 signaling mechanism are attenuated upon LIFR inhibition in-vivo.

Response: We have now included STAT3 IHC data using tumor tissue collected from in vivo studies. Results showed that HDACi treatment enhanced phosphorylation of STAT3 and treatment with HDACi+EC359 attenuated phospho-STAT3 in xenograft tumors.

4. Authors mentioned about the methodology for Mammosphere assay. However, no such data was represented in the manuscript. Please clarify.

Response: Mammosphere data was included as part of Extended data Fig. 5.

5. Abbreviation of LIFR in the abstract would help the readers.

Response: We have added this information.

Reviewer #3 :

1. This is a very impressive study characterizing anti-TNBC action of HDACi in combination with EC359. The results of cell-based studies are very convincing and clearly point at synergistic effect of EC359 with HDACi. Evaluation in xenograft with MDC-MB-468 and MDA-MB-231 cells as well PDX animal models strengthen the benefit of the combination. Gene expression and signaling analysis enhance understanding of the mechanistic events underlying the action of the compounds.

Response: We thank reviewer for these positive comments

2. The authors already characterized the activity of EC359 against GR and ER in their previous publication. This compound, however, is incredibly similar to other steroids acting as ligands/agonists/antagonists/blockers for a variety of steroid hormone receptors. One cannot exclude that EC359 is capable of both binding to LIFR and modulation of steroid hormone receptor(s). It would very useful to confirm that the results observed are not a result of steroid hormone receptor modulation and LIFR/LIF interaction blocking using a wider panel of steroid hormone receptors. The authors already tried to do it in this manuscript and in the paper where EC359 was reported. Unfortunately, STAT3, AKT, and mTOR seem to be too general to attribute to LIFR signaling only and also have well-known crosstalk with multiple steroid hormone receptors.

Response: In our previous study, we did characterize the binding activity of ER, PR and GR. The ER and GR binding of EC359 found to be nil and PR binding is only 5.9% when compared to Mifepristone (RU486) as 100%. The specific binding of EC359 towards LIFR and blocking LIF recruitment was already tested using SPR and MST assays in our earlier publication (PMID:31142661). Further, EC359 has no or limited activity in ER+BC cells, however showed potent activity in TNBC models suggesting EC359 effects are independent of steroid receptors such as ER and PR. Among our library of compounds, only those compounds having 17-alpha difluoroacetylenic group as well as lipophilic substituents at C-11 of the steroidal core showed

Extended Data Fig. 5. MDA-MB-468 xenograft tumors were treated with vehicle or EC359 or Vorinostat or combination and p-STAT3(Y705) was determined by immunostaining. Quantitation of staining was done using Image J software. Error bars represent SD. p-value was calculated using one-way ANOVA. ** p<0.01, ****p<0.0001

binding towards LIFR. Among these class of compounds, EC359 was selected based on SAR studies.

3. The authors already have LIFR-KO cells that can be treated with a combination of EC359 and HDACi. If LIFR is indeed the only target of EC359, the effect of the combination should be indistinguishable from that of HDACi only.

Response: We agree with reviewer suggestion. Our initial published studies in deed confirmed this prediction. In our initial study (PMID:31142661), we did confirm whether alteration in downstream signaling seen upon EC359 treatment such as STAT3 occurs in the cell line that has a deletion of the LIFR. Results showed that KO of LIFR significantly reduced the STAT3 activation. Furthermore, stimulation of LIFR KO cells with LIF did not activated STAT3 in this model. However, EC359 is able to block LIF-mediated STAT3 activation in LIFR-expressing control cells. These results confirm that the downstream effects seen in EC359 are due to its effects on LIFR and that STAT3 is a downstream effector of LIFR in TNBC cells. In the present study, we also found that HDACi only activated STAT3 in LIFR WT cells and HDACi failed to activate STAT3 in LIFR KO cells (Fig. 1d, Data included in the R1.2 comment above).

3. The concentrations of HDACi are very high. Why? This was not explained.

Response: We have used the HDACi concentration based on the earlier studies and also based on half maximal inhibitory concentrations (IC_{50}) in our initial studies. In the published study, the IC_{50} s of SAHA in seven out of ten hematologic tumor cell lines were below 1 μ M, while those in 25 solid tumor cells were significantly higher, up to >10 μ M (PMID: 27622335). We have added thin information in the text.

4. Error bars should be shown as SD, not SEM. SD is a measure of variability, which is the most relevant metric in these experiments.

Response: We have corrected all the *in vitro* data using SD as a measure of variability.

5. The authors keep using “therapies” when describing EC359 and HDACi and "therapeutic efficacy" when describing experiments with cells in culture. This use of “therapies” or “therapeutic efficacy” is incorrect. EC359 or HDACi are not a therapy in this context and cells are not animals or patients. The use of “efficacy” is generally incorrect when used for experiments in cells in culture. The authors measure biological effect in cells in culture as a function of HDACi concentration, which is potency. In those cases, the words “efficacy” should be replaced with “potency”.

Response: Thanks for the suggestion. We have now corrected in the text.

In summary, we are very appreciative of all the insightful suggestions by the editor and all three reviewers. We believe that we have addressed and incorporated majority of the suggestions by performing new experiments and by adding appropriate changes in the text. As can be seen from the extensive data and by the positive comments by all reviewers, this study findings discovered that Histone deacetylase inhibitors (HDACi) and LIFR inhibitor induces synergistic cell death in TNBC and these findings will be of broad interest to the readership of Communication Biology. Hence, sincerely request you to consider this manuscript for publication in Communications Biology.

Sincerely,

Ratna K Vadlamudi, Ph.D.

REVIEWERS' COMMENTS:

Reviewer #1 (Remarks to the Author):

I am satisfied with the authors' response to my previous comments and favor of the acceptance of this research manuscript by Communications Biology.

Reviewer #2 (Remarks to the Author):

The authors adequately addressed the concerns raised